# Locally Adapted and Organically Grown Landrace and Ancient Spring Cereals—A Unique Source of Minerals in the Human Diet

**DOI:** 10.3390/foods10020393

**Published:** 2021-02-11

**Authors:** Eva Johansson, Maria Luisa Prieto-Linde, Hans Larsson

**Affiliations:** Department of Plant Breeding, Swedish University of Agricultural Sciences, P. O. Box 101, 230 53 Alnarp, Sweden; maria.luisa.prieto-linde@slu.se (M.L.P.-L.); hans.larsson@slu.se (H.L.)

**Keywords:** landrace wheat, naked barley, New Nordic Diet, oats, rye, ancient wheat

## Abstract

Consumer interest in local and organic produce, sustainability along the production chain and food products contributing to health, are laying the foundation for local and organic-based diets using nutrient-dense food. Here, we evaluated 25 locally adapted landrace and ancient spring cereal genotypes per location over four locations and three years, for mineral content, nutritional yield and nutrient density. The results showed a large variation in minerals content and composition in the genotypes, but also over cultivation locations, cultivation years and for genotype groups. Highest minerals content was found in oats, while highest content of Zn and Fe was found in ancient wheats. The wheat Diamant brun, the wheat landrace Öland and naked barley showed high mineral values and high content of Zn and Fe when grown in Alnarp. Nutritional yield, of the cereals evaluated here, was high related to values reported internationally but lower than those found in a comparable winter wheat material. The nutrient density was generally high; less than 350 g was needed if any of the evaluated genotype groups were to be used in the daily diet to reach the recommended value of Zn and Fe, while if the suggested Novel Nordic Diet mix was used, only 250 g were needed. A transfer from currently consumed cereals to those in the present study, along the New Nordic Diet path, showed their potential to contribute as sustainable and nutrient-rich sources in the human diet.

## 1. Introduction

Cereals include the major crops grown throughout the globe, grown in principal in all types of climates, with wheat, rice and maize as the outranging largest crops all over [1,2,3]. In the Nordic hemisphere, the dominating cereals are wheat, barley, oats and rye [4]. Cereals in the Nordic hemisphere, can be grown either as spring or winter types, where the winter types are sown in the autumn, have to survive winter, thereby setting flowers and seeds the following summer, while spring types are sown in the spring and harvested in the coming autumn [5]. Spring cereals are primarily grown due to harsh winter conditions effecting negatively the winter cultivation of cereals [5], although spring types of cereals are also known to hold specific quality attributes, making them of interest for cultivation. Thus, spring wheat sown in the Nordic hemisphere, is known as having superior baking quality/gluten strength as compared to winter wheat [6].

Consumers are increasingly interested in both health related and sustainable aspects of food production and consumption with recent trends focusing on organic products, local production, vegetable/vegetarian/vegan food alternatives etc. [7,8,9,10,11,12]. Ancient/old cereals e.g., wheat, organically produced have also been indicated to contribute to high contents of health-related compounds (e.g., minerals, phytochemicals), high nutrient density and high mineral nutritional yield as compared to conventionally produced cereals [8,13,14,15,16,17].

Minerals are known as macro (e.g., calcium (Ca), magnesium (Mg) and potassium (K)) and micro minerals or trace elements (e.g., copper (Cu), zinc (Zn), iron (Fe), boron (B), selenium (Se)) [18], where specifically Zn and Fe deficiency have been reported as a major problem in the human diet [8]. Due to the fact that cereals are a major part of the human diet, high content of minerals and especially of Zn and Fe in cereals has the potential to contribute substantially to human health and well-being [8,13,19,20,21]. Thus, minerals content in cereals and cereal-based food have been evaluated in a range of studies [22,23,24,25,26,27]. Furthermore, large international breeding programs, such as the CIMMYT spring wheat program, has focused on biofortification breeding to combine high yield and wide adaptation with high grain Zn content and processing quality [28,29,30,31]. Content of minerals in the cereals is dependent both of the genotypic background of the cultivar and of the cultivation conditions, e.g., soil, climate and management practices [32]. The majority of studies until now have focused on winter wheat (as a major crop) from conventional cultivation systems (as being dominating), although a higher content of the majority of the minerals, including Fe and Zn, has been indicated for spring than for winter and in organic than in conventionally grown wheat [13]. Generally, limited attention has been paid towards the Nordic hemisphere cereals (barley, oats and rye), spring types of cereals, locally adapted cereals and those grown under organic conditions, despite the increasing consumer interest in local and organic production [9,10,11,12]. A comparison of mineral contents in all old Nordic spring cereals (including spring rye and naked barley) grown until the 1950s is basically lacking. Similar content of minerals in oats, barley and wheat grains have been reported [33] and levels similar to feed tables in conventionally genotypes of winter types of barely, maize, oats, rye, triticale and wheat grown in one location [34]. Recent studies have evaluated the nutritional value of “the New Nordic Diet (NND)”, with a high content of the Nordic cereals (e.g., rye bread and oatmeal), but also of other food components, such as increased levels of plant foods, foods from seas and lakes and food from the wild [35,36]. NND consumption has been shown to decrease mortality rates in Danes [37], and improve blood lipid profile and insulin sensitivity, thereby protecting against cardiovascular diseases [38].

The aim of the present study was to evaluate minerals content in locally adapted landraces and ancient genotypes of spring cereals grown organically in various locations in Sweden. Options of these genotypes as a sustainable and nutrient-rich source of minerals was evaluated and is discussed. The aim was also to evaluate nutrient density and nutritional yield of the genotypes and evaluate the contribution of consumption of these genotypes in the human diet.

## 2. Materials and Methods

### 2.1. Spring Cereal Material Produced and Used for the Study

For the present study, a total of 25 landrace and ancient spring cereal genotypes (Table 1) were grown organically in a completely randomized design with two replications in each of four locations (Ekhaga, Krusenberg, Gotland, Alnarp) and three years (2011–2013) for the present study. Due to the fact that ancient and landrace cereals are not adapted to modern high input cultivation conditions, e.g., high input often results in lodging, such conditions could not be used as a control and were therefore omitted. The spring cereal genotypes were selected so that 12 of the genotypes were the same at all the locations and years, and represented each a different type of spring cereal, i.e., rye, white oats, black oats, hulless oats, barley, hulless 2 row barley, naked 6 row barley, emmer wheat, spelt wheat, landrace wheat, old wheat cultivars (before 1950), later cultivars (1950–1960). The additional 13 genotypes per location were landraces and cultivars of barley oats and wheat locally adapted to that specific locality, with a long history and use in the Nordic climate. The overshadowing problem for spring cereals is a rainfall deficit in May–July. The selection of the plant material was done in order both to have common genotypes to allow comparisons of localities and years, but to also secure opportunities to study and understand local adaptation and what that means for mineral nutritional aspects. The cultivation locations have all been under organic conditions since 2001 and are thoroughly spread over the major cereal production areas of Sweden; Ekhaga (59°49′57″ N, 17°48′58″ E), Krusenberg (59°44′8″ N, 17°38′58″ E), Gotland (57°35’52″ N, 18°26′50″ E), Alnarp (55°39′27″ N, 13°04′51″ E), with different soil characteristics (Table 2). No weed control or fertilizer applications have been used in the trials, with the exception of at Gotland, where farmyard manure was applied in the crop rotation. Planting density used was 200 kg ha^−1^, plot size was 24 m^2^ with a harvest size of 23 m^2^. Total yield was calculated as kg ha^−1^ based on the harvest in grams from the 23 m^2^. Grain protein concentration was calculated from nitrogen determination on dried samples, applying the Dumas method on a Flash 2000 NC Analyzer (Thermo Scientific^TM^, Waltham, MA, USA), and using the conversion factor 5.7. Following the methodology adopted in Hussain et al. [13], each genotype evaluated in the present study was divided into one of the following genotype groups; Ancient (emmer and spelt wheat), Barley, Landrace wheat (landraces), Naked barley (hulless barley), Oats (black, white and hulless oats), Rye, Wheat (cultivars).

### 2.2. Mineral Analyses

Mineral analyses was carried out according to Hussain et al. [13]. Thus, about 12 g of each grain sample (whole grain) was milled for 2 × 10 s in a laboratory mill (Yellow line, A10, IKA-Werke, Staufen, Germany). Thereafter, the flour samples were stored at −20 °C until drying and digestion. The samples were dried at 40 °C for 24 h in an oven, and 0.5 g of the dried flour was digested with 10 mL of concentrated nitric acid in a microwave (MARS 5, CEM Corporation, Mathews, NC, USA). The digested samples were diluted with pure Milli Q water to 100 mL before analysis.

Then, an Inductively Coupled Plasma Atomic Emission Spectrometer (ICP-OES; OPTIMA 8300, Perkin-Elmer, Waltham, MA, USA) was used to evaluate mineral contents of Ca, Cu, Fe, K, Mg, Mn, Na, P, S and Zn. The mineral contents were calculated as absolute concentration mg/kg. Standards used in the analysis were atomic spectrometry standards from Perkin-Elmer, SPEX, AccuStandard and Merck. Calibration of the ICP-OES instrument was done by using a mixed multicomponent standard at three concentrations within the factor of 50 and calibration was maintained with independent standards. The detection limit used was three times the standard deviation based on multiple determination of the blanks treated as the sample, were blanks were treated identically and together with the samples.

### 2.3. Statistical Analyses

Analysis of variance (ANOVA), general linear model analyses (GLM), Pearson correlation analyses and principle component analysis (PCA) were carried out using the statistical analysis system (SAS; SAS Institute, Cary, NC, USA, 1985). Mean values were calculated following the ANOVA and GLM analyses, separating the means by the use of Tukey post-hoc test at *p* < 0.05. Percentage of explanation (obtained from the coefficient of determination—R^2^) of different sources (cultivar, location and their combinations) on content of total and various types of minerals were calculated using a simple linear regression analysis, following the procedure described in previous investigations [14,40,41,42]. Values of genotypes, localities, years and groups were ranked based on mean minerals content following in accordance with procedures reported previously [41]. By linear regression analysis, an independent variable can be used to predict the value of a dependent variable, and the R^2^ value of a linear regression analysis predicts how well a feature (independent variable) can explain a target (dependent variable) [43]. Thus, an R^2^ close to 1 means that the proportion is high that the independent variable explains the dependent variable (https://www.colby.edu/biology/BI17x/regression.html, access date 18 December 2020), which makes it possible to select the sources that are of highest importance to determine different traits [43]. Principal component analyses (PCA) was carried out for all minerals across all locations and years for the 12 genotypes in common for all localites to compare the effects of genotype, the locality and the year on minerals composition. Furthermore, PCA was carried out separately for all genotypes across years for each of the localities, to evaluate effects of genotypes and years on mineral composition at various locations and options for local adaptation.

### 2.4. Calculation of Nutritional Yield and Nutrient Density

Nutritional yield (NY) and nutrient density (ND) were calculated according to Morreira-Ascarrunz et al. [14], using equations below:(1)$$NY=\frac{Y\times MC}{DRI\times 365}$$
where *Y* = Yield (kg ha^−1^), *MC* = mineral content of a specific element (mg kg^−1^), *DRI* = Daily Recommended Intake of a specific element, 365 = number of days in a year.
(2)$$ND=\frac{1000\times DRI}{MC}$$
where *DRI* = Daily Recommended Intake of a specific element, *MC* = mineral content of a specific element (mg kg^−1^).

Thus, the nutritional yield is describing the number of adults that can fulfill 100% of their daily recommended intake needs with one hectare of cereals per year. For nutritional yield calculations on hulled genotypes such as the ancient wheat, oats and barley a yield correction of 75% was used following the methodology of Morreira-Ascarrunz et al. [14]. Values used for Fe (12 mg day^−1^) and Zn (8 mg day^−1^) daily recommended intake (DRI) follows the one calculated in Morreira-Ascarrunz et al. [14] and are based on an intake sufficient for the needs of 97% of individuals in an age- and sex-specific group, averaged among adult men and women.

## 3. Results

### 3.1. Variation of Mineral Contents in the Material

Generally a large variation was found in minerals content in the evaluated spring cereals (Appendix A). Mean values over three years varied 1.5 (Mg)- to 15 (Mn)-fold among genotypes and locations; Zn 29.5–56.5 mg kg^−1^, S 1233–2357 mg kg^−1^, P 3551–6027 mg kg^−1^, Na 9.7–55.5 mg kg^−1^, Mn 4.2–60.5 mg kg^−1^, Mg 1042–1732 mg kg^−1^, K 3169–4639 mg kg^−1^, Fe 27.1–64.9 mg kg^−1^, Cu 3.45–8.17 mg kg^−1^ and Ca 292–920 mg kg^−1^. General linear model analyses for the 12 genotypes in common for all localities, showed significant impact of genotype (Ge), cultivation location (C), cultivation year (Y) and C × Y interactions on content of practically all minerals while the effects of Ge × L and Ge × Y interactions were limited (Table 3). Anova analyses on the seven genotype groups (Gr) showed a significant impact of all sources (Gr, L, Y) and their interactions (Gr × L, Gr × Y, L × Y) on in practical content of all minerals.

R square values used to explain the percentage of explanation for the different minerals, showed for most of the minerals a higher degree of explanation for the genotype/genotype group analyzed as compared to the location and year used for cultivation. However, for content of Zn, Mn and Cu, cultivation locality turned out to have the highest degree of explanation and also for Fe, the locality contributed to a high degree of explanation (Table 4). Furthermore, cultivation year showed a high impact on the degree of explanation for several of the analyzed minerals (e.g., Zn, K, Cu, Ca; Table 4).

### 3.2. Minerals Variation by Genotype, Cultivation Location, Cultivation Year and Genotype Group

The comparison of 12 spring cereal genotypes over the four environments and years, showed clear differences in mineral content and composition among the genotypes (Table 5). High level of S, P, Mg and Fe was found in black oats cultivar Engelbrekt, high level of Zn was found in Emmer Gotland. The spring barley variety Ingrid showed low levels of Zn, S, P, Mn, Mg, K, Fe Cu and Ca. Among the localities, high levels of Zn, S, Fe and Cu were found in cereals grown at Ekhaga, while low levels of Zn, S, Mg, Fe and Cu were found in cereals grown in Alnarp (Table 5). High levels of Zn, Na, K and Ca were found in the cereals when grown 2013, while high levels of Mn, K and Cu were found for those grown 2011 (Table 5). Sorting the spring cereals investigated into different groups, showed ancient wheat such as spelt and emmer to contain high levels of Zn, Mg and Cu, while oats showed high levels of S, P, Mn and Fe. Low levels of most of the minerals were found in barley (Table 5).

### 3.3. Combined Impact of Genotype/Genotype Group, Cultivation Location and Year on Minerals Content

The score plot from the principal component analyses (PCA) of genotypes, locations and years (Appendix A) and genotype groups, locations and years (Figure 1a), indicated a co-variation of the minerals over the different samples, with positive values on the first principal component (PC1) for all the evaluated minerals, also verified by highly significant Pearson correlation coefficients between most of the minerals (Appendix A).

The loading plot of the PCA of the twelve spring cereal samples differentiated a spread of the samples based on cultivation locality along PC1, with the majority of the Ekhaga samples showing positive PC1 values, thereby indicating high mineral levels and the majority of the Alnarp samples with negative PC1 values indicating low mineral levels (Appendix A). However, the spring barley cultivar Ingrid was consistently over all cultivation locations found with low PCA2 values indicating low levels of minerals. For the rest of the 12 genotypes, no consistent pattern could be differentiated with the PCA explaining their variation in minerals content and composition (Appendix A).

The loading plot of the PCA of the seven genotype groups clearly differentiatedthe different genotype groups. Thus, the majority (11 out of 12) of the oat group samples were found with positive PC1, only the sample from Alnarp 2011 showed a negative PC1 value, thereby indicating a general high mineral content in the oats samples evaluated in the present study (Figure 1b). Furthermore, the majority of the ancient wheat group samples (9 out of 12) were found with positive PC1 values and negative PC2 values indicating generally high Fe and Zn values, although two of the Alnarp samples (2011 and 2013) were found with negative PC1values (Figure 1b). The barley genotype group samples were generally found (12 out of 12 samples) with low PC1 values, indicating a lower minerals content than what was found in the rest of the samples evaluated here (Figure 1b). The majority of the naked barley group samples were found gathered along the PC2 with positive values (9 out of 12), indicating high levels of Na and K, with only two samples (Alnarp 2011 and Gotland 2012) showing negative PC2 values (Figure 1b). The samples of the rest of the genotype groups (rye, wheat and wheat landraces showed a larger spread of PC values in the loading plot (Figure 1b).

### 3.4. Local Adaptation of the Genotypes

PCA analyses separately on samples from a specific cultivation location, verified for each of the locations a similar score plot as described above for the full material (Figure 1a). Thus, for each of the locations, all minerals clustered with positive PC1 values, indicating a co-variation of the minerals in samples for each location. Thus, genotypes with consistently positive PC1 values in a specific location, over the three years of study, should be considered as having a good chance to produce high mineral content at that location independently of yearly climate fluctuations, and such genotypes are presented in Table 6. From the present study, it was clearly shown that oats and ancient wheat was outstanding as high mineral genotypes groups across the four cultivation locations (Table 6). Among the oats, some genotypes as Engelbrekt and Virma, resulted in positive PC1 values, indicating high minerals content, across all four locations and in all evaluted years (Table 6). Such genotypes are specifically interesting for breeding and produce across different local environments across Sweden. Some of the genotypes were only tested in some of the locations (1–3) due to an expectation of a local adaptation to that locality of the genotype. Several such oat genotypes were found to perform well in the locality/ies they were tested (e.g., Bambu, Orion, Sisu, Sol etc.; Table 6), and these genotypes might be of interest to test in a broader set of environments, to verify their possible local adaptation. For landrace wheat, wheat and naked barley, a clear local adaptation was seen in the present material. Thus, positive PC1 values was obtained for Diamant brun (wheat), Hulless 2row barley (naked barley) and Öland (landrace wheat) was obtained in Alnarp, which was not seen in the other localities (Table 6), despite these genotypes belonged to the 12 grown in all four localities. These findings might indicate a local adaptation of these genotypes from the south of Sweden to cultivation environments similar to the one in Alnarp.

PCA for each locality on four selected minerals (Zn, Fe, Mg and Cu) resulted in a score plot with positive PC1 for all minerals (not shown), and depicted clearly the ancient wheats across all four locations with positive PC1, thereby whith high levels of the selected four minerals (Table 6). Furthermore, several oats genotypes in particular when grown in Krusenberg, depicting some local adaptation, and Diamant brun (wheat), Hulless 2row barley (naked barley) and Öland (landrace wheat) when grown in Alnarp, showed positive PC1 values (Table 6) indicating high levels of the minerals of choice, and local adaptation.

### 3.5. Nutritional Yield and Nutrient Density

Mean nutritional Zn and Fe yield of the various genotype groups evaluated here was, respectively, 20–30, and 27–50 adults/ha and year, with high values for wheat and landrace wheat and low values for oats and barley (Table 7). Significant differences were noted for nutrient density among the evaluated groups, with highest nutrient density (least amount needed to be consumed to obtain the daily recommended intake) found in oats and ancient wheat for the combination of Fe and Zn (Table 7). Yield of the genotype groups varied with mean values from 2330 to 3550 kg ha^−1^ and grain protein content varied also significantly with values from 11.0 to 13.3% (Table 7). A significantly negaitve Pearson correlation (*p* < 0.005) was found between yield and grain protein concentration.

## 4. Discussion

The present study clearly showed specific genotypes, of the locally adapted spring cereal landraces and ancient genotypes grown organically, with high content of minerals and high nutrient density, although the variation among genotypes, cultivation locations and years were striking with 1.5- to 15-fold differences. The high mineral content genotypes depicted contribute opportunities to select mineral-rich genotypes for local production of nutritive food alternatives to be used e.g., in the New Nordic Diet, and also for breeding of novel high-nutrient cereals. Generally, the oats genotypes showed the highest minerals content, while ancient wheat showed high content of Zn and Fe. High content of minerals were also found, e.g., in the wheat genotype Diamant brun, the wheat landrace genotype Öland and the Hulless 2row barley genotype, and in particular when grown in Alnarp.

The large differences in content of various minerals shown in the samples evaluated here correspond well with previous results [32], which have proven that the selection of a wide array of genetic material and a spread of cultivation environments will result in a large variation in any type of compound analyzed in the plant material. Previous studies have indicated cultivation location as a major contributor to the minerals content in the cereal grain, although genetic impact on the minerals content has also been described previously, as well as genotype x environment interactions [13,14,32,43,44,45]. However, the present investigation has clearly depicted the importance of the genotypes on the minerals content and composition. In the present study, covering minerals content of a wide array of Nordic traditional cereals, genotypes were outlined as the most important source for the mineral variation of all minerals except Zn and Cu, for which cultivation location showed a higher percentage of explanation. Similarly, the genotype groups showed a higher percentage of explanation than the environmental factors (location and year) for all minerals except Zn, K, Fe and Cu. In particular for K, but also to a high extent for Zn and Cu, the cultivation year showed a high percentage of explanation indicting the importance of taking also this source into consideration when evaluating minerals content in spring cereals. Thus, to secure a full understanding of the mineral variation in a broad cereal genetic material grown over several locations and years, use of multivariate statistical methods are required with potential to compare sample variation over all sources.

Previous studies [32] have indicated the width of the parameters (how broad genetic variation and how dispersed cultivation environments that are used) as the predictor for the importance of each of the parameters. Here, the genotypic material (various cereal types) might be judged as representing a broader width than the cultivation locations (spread across Sweden). Differently than in most studies, e.g., [13,14,32,45,46,47,48,49,50,51,52], the present study showed limited interactions between genotypes, cultivation location and cultivation year for the variation in minerals content and composition in the cereals evaluated. The absence of genotype by environment interactions indicate that individual that are genetically alike is expected to be homogenous independent of cultivation environment [53]. In the present investigation, the 12 genotypes being the same in all four cultivation environments are from various genotype groups with unequal number of each type. If genotypes from the same genotype group is seen as clusters, each genotype within a cluster (genotype group) can be expected to vary more similar over environments to genotypes within the same cluster than to genotypes within other clusters. Thus, the selection of the genotypic material in the present study might explain the lack of genotype x environment interactions seen for the 12 genotypes grown and analyzed across all environments. This also explains the more common behavior with genotype x environment interactions that was found for genotype groups x environments. However, due to the unbalanced character of the genotype groups in the present study (e.g., the rye group consisted of one genotype while oats and wheat consisted of 6–8 genotypes per location), multivariate statistical methods and treatments to overcome such issues are a necessity.

From the large variation in the content of different minerals depicted in the present study, specifically high contents of minerals were found both in certain cereal groups and in specific genotypes. Thus, the oat genotypes were here, generally found with a high content of minerals, and high content of Zn, Fe, Mg and Cu were generally noted for the ancient wheat genotypes across all four cultivation environments and years. Among the minerals, Fe and Zn are considered the most important in relation to human health [8,43]. A high content of Zn and Fe in ancient wheat grown in Sweden has also been reported in previous publications [13,14]. The levels reported here in ancient spring genotypes are somewhat lower as compared to values reported for winter wheat grown during the same period and in the same locations [14] but somewhat higher than in a mixed material of spring and winter genotypes grown during a somewhat earlier period of time [14]. High levels of Fe and Zn has consistently been reported in wild emmer wheat genotypes [43], and in diploid wheat, quantitative traits loci (QTL) for Zn and Fe content have been mapped to chromosomes 2A and 7A [44]. Thus, the findings in the present study of high contents of Fe and Zn in Nordic ancient spring wheat genotypes correspond well with previous findings on a broad range of ancient wheat.

Oats have in previous studies been identified as a functional food, due to its high content of soluble fiber, lipids, proteins, vitamins, minerals and phytochemicals such as polyphenols [54]. A recent study on mineral content in white oat genotypes from Brazil, reported a wide range of minerals content, with Fe values from 38 to 63 mg/kg and Zn values from 27 to 67 mg/kg [54] which also correspond to the variation in earlier studies [34,55,56]. Thus, the oats genotypes evaluated here did not outperform the best oat genotypes determined in previous studies for Zn and Fe content. Most of the oat genotypes in the present study did neither show the highest Zn and Fe values among samples evaluated here, although many of the oat genotypes evaluated showed generally high and stable minerals content, including all 10 minerals evaluated and across all cultivation locations and years.

The barley genotypes evaluated in the present study, consistently showed the lowest mineral contents among the samples. However, most previous investigation report even lower values for barley and also for rye, for content of Zn, Fe and most other minerals [55], than reported in this study. Furthermore, the naked barley genotypes in the present study, showed generally higher mineral contents than was found for the barley genotypes, with especially high levels of Na. Mineral contents in naked barley from other studies are lacking. The differences in mineral content between barley and naked barley might be explained by the pearling of the barley, which is not necessary for naked barley.

The present study was also able to pin point high performing genotypes as well as high performing genotype groups across all cultivation location and years and those with more local adaptation, performing well in a certain cultivation location. Thus, the oat genotypes Engelbrekt (black oat) and Virma (white oat) showed high mineral levels across all cultivation location and the ancient wheats Emmer Gotland and Spelt wheat Gotland showed high Zn ad Fe content across all locations. The wheat genotype Diamant brun, the wheat landrace Öland and the hulless 2row barley were among those genotypes performing well in a certain location, e.g., Alnarp. Previous studies have shown variants of the landrace Öland with high mineral content and high content of carotenoids [13,15]. An adaptation over a wide array of environments is favorable for traditional breeding of high value characters in approved cultivars although local adaptation might be favorable for local production of high value genotypes for certain products.

Corresponding with previous results [13], the cultivation location Ekhaga contributed a higher mineral content, especially of Zn, Fe, Cu, Ca and S to the samples. The major difference in the present study, of Ekhaga compared to the other localities was a difference in K-Al, with significantly higher values in Ekhaga, which was also the case in the previous study carried out on wheat during 2001–2007 [13]. The K-Al value in the soil describes the availability of K in the soil for the plant, which may enhance the minerals uptake.

The nutritional yield (e.g., Zn NY for wheat = 28, Fe NY for wheat = 50) of the spring genotypes evaluated here was generally lower, specifically for Zn, as compared to corresponding values for winter wheat (Zn NY= 46–52, Fe NY = 37–54) reported earlier [14]. These differences can easily be explained by the generally lower yield always found for spring as compared to winter wheat when grown in the Nordic countries. Due to the low yield, winter cereals are often preferred by the grower but spring cereals can be an alternative due to harsh winter conditions hampering the winter cereal cultivation and also for the specific quality attributes that can be found in spring cereals, e.g., improved baking quality in wheat [5,6]. However, the yield difference between winter and spring wheat, makes comparisons of nutritional yield rather unbalanced. Studies on nutritional yield in spring wheat is currently lacking, and is reported for the first time for Nordic spring cereals. Furthermore, comparing the nutritional yield of the spring cereals in the present study with nutritional yield in cereals in an international context clearly shows the potential of the genotypes presented here. Thus, international data on the nutritional yield reports both Zn NY and Fe NY of 20–25 adults/ha and year for oats, 5–7 for wheat and 15–20 for barley and rye [57], and the corresponding numbers of the material presented here are 22–28 for oats, 28–50 for wheat and 21–37 for barley and rye. The differences in nutritional yield can be explained by differences in the genetic material and the cultivation conditions where organic cultivation of traditional genotypes might contribute to high values [10], while total yield (e.g., varying largely in wheat production) is also a contributing factor in the calculation of nutritional yield.

Similarly to has been described for traditional and organically produced winter wheat [14], the present study on traditional and organically produced spring wheat, showed favorable nutrient density in the wheat samples as compared to conventionally produced wheat [58,59]. Here, comparable nutrient densities to wheat were also shown for the rest of the genotype groups evaluated, with the exception of barley. Thus, locally adapted and organically produced cereals is a highly nutritional worthy alternative for the Nordic countries.

Cereals are the major staple for a large proportion of the worlds’ population, contributing more than 40% of the daily calories and protein needed by the human population [1]. Due to the high intake, a shift towards cereals which are more highly nutritious and nutrient dense than those currently consumed will contribute a major change for human health. The present study verifies the opportunity to select and cultivate locally adapted and sustainably (organically or low-input) cultivated cereals with high minerals content and nutrient density that would benefit human health through its nutritional value.

Combining the use of high nutrient density cereals with a shift to a healthy diet such as the “Mediterranean” or the “Novel Nordic Diet (NND)” would add additional nutritional health to the human population. The NND suggests a transition to an increased amount of local and organically produced whole grain cereals in the diet for increased sustainability combined with improved human health [60]. Recent reports clearly show a change in the dietary intake in the Swedish population [61]. Less raw products such as flour, potatoes, sugar etc. are consumed and instead higher processed products are prioritized. Also, consumption of rice and pasta have increased on the cost of primarily potatoes. Table 8 shows a comparison of the contribution of minerals from 250 g (which is enough for the coverage of the daily requirements for most of the genotype groups of the present study) of different types of products, including the wheat genotypes analyzed here. In addition, we have included one product defined here as the NND mix. The NND suggests a transition towards whole grain products, to local production and to a consumption of less refined products [60]. The NND will also gain nutritional importance by the use of a combination of cereals. The oats, the ancient wheat, and some genotypes of wheat, landrace wheat and naked barley were found here as the most mineral rich and dense genotypes/genotype groups. However, other studies to come will probably show other important nutritional characters in genotypes other than those being the most mineral rich/dense.

## 5. Conclusions

Mineral content and composition vary considerably in locally adapted and organically grown landraces and ancient genotypes of spring cereals, with clearly high values in certain genotypes and genotype groups. The genotypes and genotype groups were of significant importance to determine the content and composition of the minerals in the grain. However, to determine the content of certain minerals, e.g., Zn and Cu, cultivation location played a larger role than the genotypes. In addition, the cultivation year was of significance for the content of some minerals, e.g., K and Cu. The oats showed in general the highest mineral content although ancient wheats showed the highest content of Zn and Fe. Specific genotypes, e.g., the wheat genotype Diamant brun, the landrace wheat genotype Öland and the hulless 2row barley genotype was found with high mineral as well as Zn and Fe content when grown in Alnarp, due to the local climate of the most Southern cultivation location included, including cultivation temperature, precipitation, day-length and soil conditions. Although lower that in comparable winter wheat genotypes grown at the same locations, the nutritional yield of the spring cereals were generally high in an international context. Furthermore, all evaluated genotype groups, with the exception of barley, showed a high nutrient density. The use of either e.g., the wheat grains or a mix of genotypes from the different genotype groups in the New Nordic Diet concept, will contribute more or less the daily requirement of the most important minerals (Zn, Fe, Cu, Mg), by the consumption of 250 g day^−1^. Thus, as a source of mineral nutrition, the spring cereals evaluated here, outperform most other popular food products, which they have the potential to replace, such as rice, pasta, white flour, whole grains from conventional production. A mix in the food of the different genotype groups evaluated is suggested to secure also other nutritive compounds beside the minerals, and whole grain and less processed products should be taken into consideration, following the NND recommendations to secure presence and availability of the minerals.

## Figures and Tables

**Figure 1 foods-10-00393-f001:**
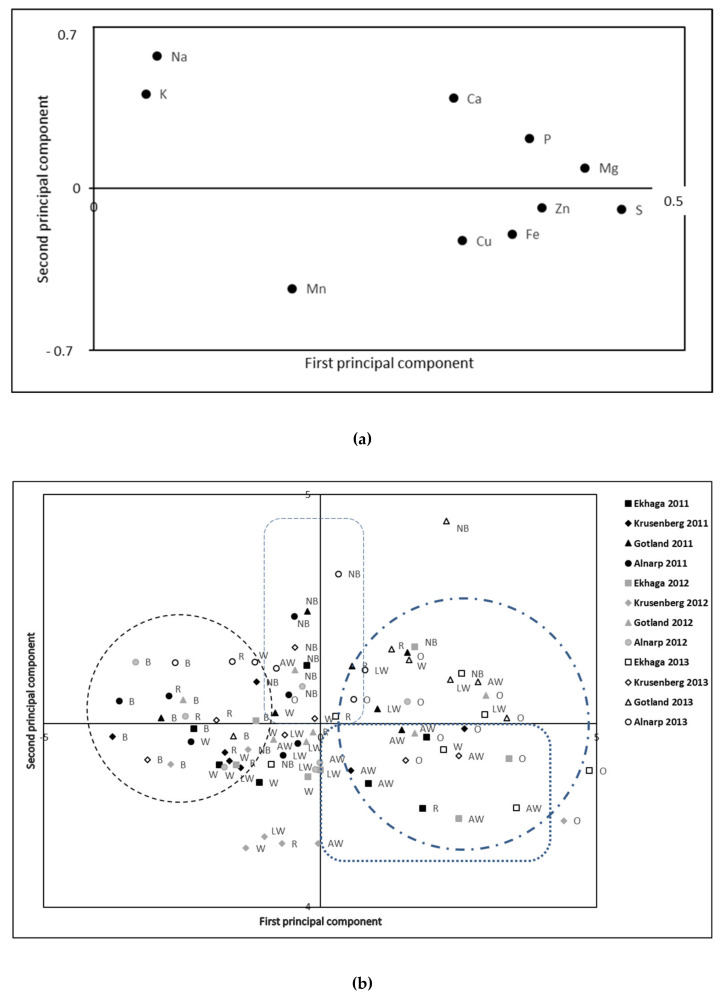
Score (**a**) and loading (**b**) plot from principal component analysis of mineral content in spring cereal genotype groups (Rye-R, Oats-O, Wheat-W, Ancient wheat-AW, Naked barley-NB, Barley-B, Landrace wheat-LW) grown at different localities (Ekhaga, Krusenberg, Gotland, Alnarp) during different years (2011, 2012, 2013). First principal component explained 38.4% of the variation and the second principal component explained 17.7% of the variation. Black dotted circle to the left appoint the majority of the Barley (B) group samples, black dotted rectangle in the top indicate the majority of the Naked barley (NB) group samples, blue dotted circle to the right mark majority of the Oat (O) group samples, and low dotted blue rectangle mark the majority of the Ancient wheat (AW) group samples.

**Table 1 foods-10-00393-t001:** Name, type, origin and place of cultivation of genotypes used in the present study.

Genotype	Type	Origin	Place ^1^
Algot	Wheat	Cultivar 1953	E, K, A
Alva	Barley	Cultivar 1977	A
Argus	Black oats	Cultivar 1926	G
Atle	Wheat	Cultivar 1953	G A
Atson	Wheat	Cultivar 1954	A
Aurore	Wheat	Cultivar 1929	K
Balder	Barley	Cultivar 1945	E, K
Bambu	White oats	Cultivar 1934	E
Blenda	White oats	Cultivar 1950	G
Dacke	Wheat	Cultivar 1990	E, K
Diamant brun	Wheat	Cultivar 1928	E, K, G, A
Domen	Barley	Cultivar 1959	E
Dragon	Wheat	Cultivar 1988	G, A
Ella	Wheat	Cultivar 1950	E, K, G, A
Emmer Gotland	Emmer wheat	Primitive	E, K, G, A
Engelbrekt	Black oats	Cultivar 1924	E, K, G, A
Extra Klock	Black oats	Cultivar 1955	K
Gotlandskorn	Barley	Cultivar 1915	G, A
Gullkorn	Barley	Cultivar 1913	G
Hulless 6row barley	Hulless barley	Genebank	E, K, G, A
Hulless 2row barley	Hulless barley	Genebank	E, K, G, A
Hulless oats	Hulless oats	Genebank	E, K, G, A
Ingrid	Barley	Cultivar 1958	E, K, G, A
Jusso	Rye	Landrace	E, K, G, A
Kajsa	Barley	Cultivar 1977	E, K
Klock	Black oats	Cultivar 1917	A
Kärn	Wheat	Cultivar 1946	E, K
Landrace Dalarna	Wheat	Landrace	E, K, G
Landrace Halland	Wheat	Landrace	E, K
Lina	Barley	Cultivar 1982	A
Orion	Black oats	Cultivar 1920	E, K, G
Osmo	Back oats	Cultivar 1921	A
Palu	White oats	Cultivar 1945	A
Prins	Wheat	Cultivar 1965	E, G
Rika	Barley	Cultivar 1949	G
Seger	White oats	Cultivar 1908	K, A
Selma	White oats	Cultivar 1970	A
Sisu	White oats	Cultivar 1953	E
Sol	White oats	Cultivar 1950	E, K
Spelt wheat Gotland	Spelt wheat	Spelt	E, K, G, A
Spelt wheat Gotland d	Spelt wheat	Spelt	G
Summer oats	White oats	Landrace	G
Svanhals	Barley	Cultivar 1903	K
Ur Gotland	Black oats	Landrace	G
Walter	Wheat	Cultivar 1972	A
Virma	White oats	Cultivar 1988	E, K, G, A
Öland	Wheat	Landrace	E, K, G, A

^1^ E = Ekhaga, K = Krusenberg, G = Gotland, A = Alnarp.

**Table 2 foods-10-00393-t002:** General characteristics of soil at different locations.

Location	pH ^a^	Organic Matter (%)	Clay (%)	P-Al ^b^ (mg 100 g^−1^)	K-Al ^b^ (mg 100 g^−1^)	FYM ^c^	Organic Since
Ekhaga	5.7–6.1	7.2–9.8	35–38	6.2–7.1	25–27.8	No	1987
Krusenberg ^d^	5.8	2.0	9.0	8.0	5.0	No	2001
Gotland	7.5–8.3	2.5–3.9	18–20	5.2–9.8	9.1–10.4	Applied	1987
Alnarp	7.3–7.8	3.1–4.5	18–22	7.7–26.7	10.2–18.7	No	1992

^a^ from soil-water sample; ^b^ P-Al and K-Al methods used [39]; ^c^ Farm yard manure; ^d^ Only one soil sample was analyzed from this location.

**Table 3 foods-10-00393-t003:** Mean squares from the general linear model analyses (GLM) for 12 genotypes and analyses of variance (ANOVA) for seven genotype groups indicating impact on minerals content from genotypes (Ge), Genotype groups (Gr), localities (L), years (Y) and their interactions.

Source	Df	Zn (10^3^)	S (10^5^)	P (10^6^)	Na (10^3^)	Mn (10^3^)	Mg (10^5^)	K (10^6^)	Fe (10^2^)	Cu	Ca (10^5^)
12 genotypes over four locations and three years											
Genotype	11	0.32 ***	4.57 ***	2.57 ***	2.32 ***	1.07 ***	1.80 ***	1.12 ***	2.13 ***	2.34 ***	1.47 ***
Locality	3	1.18 ***	4.82 ***	1.19 ***	1.09 ***	4.16 ***	1.20 ***	0.06	6.72 ***	22.7 ***	0.28 ***
Year	2	0.08 ***	2.32 *	0.10	1.92 ***	0.34 ***	0.70 ***	4.26 ***	1.57 ***	21.7 ***	2.47 ***
Ge*L	33	0.03	0.20	0.13	0.21	0.10 ***	0.15 *	0.08	0.54	0.36	0.03
Ge*Y	21	0.03	0.23	0.21 **	0.36 ***	0.07 *	0.13	0.10 *	0.60	0.40	0.12 ***
L*Y	6	0.14 ***	2.03 ***	0.66 ***	0.82 ***	0.17 ***	0.31 ***	0.31 ***	3.12 ***	4.01 ***	0.07 ***
Error	182	0.02	0.17	0.09	0.14	0.03	0.08	0.05	0.56	0.24	0.01
7 genotype groups over four locations and three years											
Group	6	0.79 ***	31.4 ***	15.1 ***	5.33 ***	4.84 ***	8.82 ***	1.68 ***	10.3 ***	6.98 ***	6.26 ***
Locality	3	1.93 ***	11.1 ***	3.20 ***	2.87 ***	9.84 ***	2.86 ***	0.10	9.28 ***	50.3 ***	0.85 ***
Year	2	2.33 ***	4.65 ***	0.73 ***	3.87 ***	1.25 ***	1.17 ***	6.44 ***	10.2 ***	54.7 ***	3.81 ***
Gr*L	18	0.09 ***	0.28	0.15	0.34 ***	0.35 ***	0.41 ***	0.34 ***	1.23 ***	0.93 ***	0.10 ***
Gr*Y	12	0.06 ***	0.54 ***	0.48 ***	1.14 ***	0.15 ***	0.33 ***	0.24 ***	1.61 ***	1.29 ***	0.59 ***
L*Y	6	0.21 ***	5.24 ***	1.28 ***	1.06 ***	0.44 ***	0.49 ***	0.52 ***	5.94 ***	8.66 ***	0.26 ***
Error	84	0.03	0.18	0.13	0.12	0.03	0.09	0.05	0.51	0.44	0.03

*,**, *** = Significant at *p* < 0.05, 0.01 and 0.005.

**Table 4 foods-10-00393-t004:** Percentage of explanation (obtained through the coefficient of determination [R^2^] from simple linear regression analyses) of 12 genotypes across four locations and three years, and 7 genotype groups across four locations and three years, on amount of various minerals.

Source	Zn	S	P	Na	Mn	Mg	K	Fe	Cu	Ca
Genotype	25.1	42.3	50.6	28.2	29.8	43.8	36.8	14.0	17.0	55.9
Locality	29.3	15.0	8.58	6.20	39.1	8.32	0.94	10.2	25.9	3.86
Year	9.32	1.90	0.47	7.33	2.52	1.20	19.2	2.18	22.0	15.1
Group	5.16	53.0	59.6	19.2	35.6	51.1	18.7	6.24	6.53	54.5
Locality	23.2	9.45	6.19	5.74	37.5	7.19	0.25	8.70	29.0	3.54
Year	16.1	3.14	0.46	7.11	2.44	1.64	22.9	5.79	19.0	11.1

**Table 5 foods-10-00393-t005:** Mean values of content of minerals (mg kg^−1^) in twelve genotypes over four localities and three cultivation years.

Source	Zn (10^1^)	S (10^3^)	P (10^3^)	Na (10^1^)	Mn (10^1^)	Mg (10^3^)	K (10^3^)	Fe (10^1^)	Cu	Ca (10^2^)
Genotypes										
Diamant brun	4.60 ^a,b^	1.57 ^b,c,d,e^	4.36 ^c,d^	1.91 ^b^	3.41 ^a,b^	1.30 ^c,d^	3.60 ^c^	4.65 ^a^	5.32 ^a,b,c,d^	5.57 ^d^
Ella	4.01 ^b,c^	1.50 ^d,e^	4.27 ^d^	2.11 ^b^	3.01 ^b,c^	1.30 ^c,d^	3.69 ^c^	4.08 ^a,b^	5.45 ^a,b,c,d^	4.95 ^e,f^
Emmer Gotland	4.82 ^a^	1.61 ^b,c,d^	5.00 ^a,b^	2.28 ^b^	3.09 ^b,c^	1.46 ^a,b^	4.15 ^a,b^	4.38 ^a^	6.13 ^a,b^	4.01 ^g^
Engelbrekt	3.93 ^b,c^	1.95 ^a^	5.41 ^a^	2.82 ^b^	3.78 ^a,b^	1.47 ^a^	3.65 ^c^	4.89 ^a^	5.23 ^b,c,d^	6.16 ^c^
Hulless 6row barley	3.79 ^c^	1.53 ^c,d,e^	4.57 ^b,c,d^	5.69 ^a^	1.55 ^e,f^	1.30 ^b,c^	4.22 ^a,b^	4.32 ^a,b^	4.86 ^c,d^	5.48 ^d^
Hulless 2row barley	3.79 ^c^	1.60 ^b,c,d,e^	5.03 ^a,b^	5.96 ^a^	1.64 ^d,e^	1.36 ^a,b,c^	4.34 ^a^	4.61 ^a^	5.58 ^a,b,c^	5.59 ^d^
Hulless oats	3.47 ^c,d^	1.84 ^a,b^	4.96 ^a,b,c^	2.16 ^b^	3.90 ^a,b^	1.40 ^a,b,c^	3.66 ^c^	4.30 ^a,b^	4.49 ^c,d^	7.93 ^a^
Ingrid	2.70 ^d^	1.34 ^e^	3.70 ^e^	3.26 ^b^	0.80 ^f^	1.08 ^e^	3.52 ^c^	3.09 ^b^	4.46 ^d^	3.22 ^g^
Jusso	4.17 ^a,b,c^	1.50 ^d,e^	4.22 ^d,e^	2.00 ^b^	2.40 ^c,d^	1.15 ^d,e^	4.43 ^a^	4.33 ^a,b^	5.38 ^a,b,c,d^	4.69 ^f^
Spelt wheat Gotland	4.41 ^a,b,c^	1.80 ^a,b,c^	4.90 ^a,b^	2.18 ^b^	2.86 ^a,b^	1.47 ^a^	3.72 ^b,c^	4.27 ^a^	6.76 ^a^	3.34 ^g^
Virma	3.90 ^b,c^	2.00 ^a^	5.12 ^a,b^	1.88 ^b^	4.08 ^a^	1.41 ^a,b,c^	3.89 ^b,c^	4.55 ^a^	4.97 ^c,d^	6.62 ^b^
Öland	4.15 ^a,b,c^	1.60 ^b,c,d,e^	4.68 ^b,c,d^	1.54 ^b^	3.16 ^b,c^	1.40 ^a,b,c^	3.70 ^c^	4.92 ^a^	5.27 ^a,b,c,d^	5.19 ^d,e^
Localities										
Ekhaga	4.88 ^a^	1.76 ^a^	4.68 ^b^	3.14 ^a^	3.41 ^b^	1.37 ^a^	3.87 ^a^	5.01 ^a^	6.41 ^a^	5.18 ^b^
Krusenberg	3.91 ^b^	1.60 ^b,c^	4.42 ^c^	1.82 ^b^	4.13 ^a^	1.29 ^b^	3.84 ^a^	4.27 ^b^	4.89 ^b,c^	4.71 ^c^
Gotland	4.14 ^b^	1.72 ^a,b^	4.97 ^a^	3.30 ^a^	1.43 ^d^	1.42 ^a^	3.96 ^a^	4.20 ^b^	5.30 ^b^	5.59 ^a^
Alnarp	3.23 ^c^	1.49 ^c^	4.64 ^b,c^	3.05 ^a^	2.25 ^c^	1.28 ^b^	3.94 ^a^	4.06 ^b^	4.56 ^c^	5.24 ^b^
Years										
2011	3.79 ^b^	1.60 ^a^	4.68 ^a^	2.26 ^b^	3.18 ^a^	1.31 ^a^	4.11 ^a^	4.20 ^a^	4.53 ^b^	5.17 ^b^
2012	3.76 ^b^	1.64 ^a^	4.71 ^a^	2.63 ^b^	2.61 ^b^	1.35 ^a^	3.55 ^b^	4.51 ^a^	5.83 ^a^	4.50 ^c^
2013	4.56 ^a^	1.70 ^a^	4.63 ^a^	3.80 ^a^	2.62 ^b^	1.36 ^a^	4.08 ^a^	4.49 ^a^	5.63 ^a^	6.05 ^a^
Groups										
Rye	4.17 ^a,b^	1.50 ^b,c,d^	4.22 ^d,e^	2.00 ^b,c^	2.40 ^b,c^	1.15 ^c^	4.43 ^a^	4.33 ^a,b^	5.38 ^a,b^	4.69 ^b,c^
Oats	3.98 ^b^	2.03 ^a^	5.40 ^a^	2.16 ^c^	3.84 ^a^	1.49 ^a^	3.85 ^c,d^	4.88 ^a^	5.05 ^b^	6.69 ^a^
Wheat	3.84 ^b^	1.50 ^c^	4.20 ^d,e^	2.20 ^c^	2.89 ^b^	1.28 ^b^	3.80 ^c,d^	4.02 ^b,c^	5.10 ^b^	4.94 ^b^
Ancient wheat	4.68 ^a^	1.72 ^b^	5.04 ^b^	2.20 ^c^	3.04 ^a,b^	1.49 ^a^	3.98 ^b,c^	4.34 ^a,b^	6.36 ^a^	3.74 ^c,d^
Naked barley	3.79 ^b^	1.57 ^b,c^	4.79 ^b,c^	5.82 ^a^	1.59 ^c,d^	1.33 ^b^	4.28 ^a,b^	4.46 ^a,b^	5.21 ^b^	5.54 ^b^
Barley	3.09 ^c^	1.34 ^d^	3.98 ^e^	3.50 ^b^	1.01 ^d^	1.12 ^c^	3.67 ^d^	3.54 ^c^	4.82 ^b^	3.54 ^d^
Landrace wheat	4.07 ^a,b^	1.57 ^b,c^	4.45 ^c,d^	1.98 ^c^	3.40 ^a,b^	1.32 ^b^	3.89 ^c,d^	4.38 ^a,b^	5.35 ^b^	4.97 ^b^

Numbers followed by the same letters within a column for each of the sources do not differ significantly by the use of Tukey post-hoc test at *p* < 0.05.

**Table 6 foods-10-00393-t006:** PC1 values from each cultivation location for genotypes with positive PC1 values in all years for all ten minerals evaluated and for four (Zn, Fe, Mg and Cu) minerals of specific nutritional importance [14].

Location and Genotype	Genotype Group	All Ten Minerals			Four Selected Minerals		
		2011	2012	2013	2011	2012	2013
Ekhaga							
Bambu	Oat	1.67	3.71	3.30			
Emmer Gotland	Ancient				0.05	2.60	1.93
Engelbrekt	Oat	0.10	2.80	1.76			
Orion	Oat	2.83	4.28	3.01	0.85	2.43	2.18
Sisu	Oat	0.14	2.14	2.00			
Sol	Oat	0.31	3.64	0.79			
Spelt Gotland	Ancient	0.06	1.11	X	0.03	2.38	X
Virma	Oat	1.07	3.34	2.77			
Krusenberg							
Emmer Gotland	Ancient				1.56	1.43	0.66
Engelbrekt	Oat	2.83	1.14	2.22	1.71	0.61	2.29
Extra klock	Oat	2.16	2.15	3.32	0.17	0.49	3.09
Orion	Oat	3.32	2.05	5.13	1.57	1.12	4.15
Seger	Oat	2.91	3.54	5.22	1.22	1.86	393
Sol	Oat	2.51	1.93	4.21	0.70	0.80	3.04
Spelt Gotland	Ancient				0.30	2.36	1.34
Virma	Oat	2.41	0.36	2.34			
Gotland							
Argus	Oat	0.51	3.36	X			
Blenda	Oat	0.85	1.26	1.72			
Emmer Gotland	Ancient	1.29	1.07	1.28	1.62	0.62	2.45
Engelbrekt	Oat	0.61	2.28	0.83			
Hulless oats	Oat	0.53	1.44	X			
Orion	Oat	1.52	4.20	1.68			
Spelt wheat Gotland	Ancient	1.02	0.73	X	1.61	1.29	X
Spelt wheat Gotland d	Ancient				0.08	1.21	3.52
Sommarhavre	Oat	2.45	3.08	2.39	0.02	0.86	2.91
Ur Gotland	Oat	2.98	4.17	2.50	0.31	2.07	2.28
Virma	Oat	1.13	0.86	1.69			
Alnarp							
Diamant brun	Wheat	0.72	1.72	0.88	0.81	1.99	1.12
Engelbrekt	Oat	0.67	3.75	X			
Emmer Gotland	Ancient				0.96	1.93	0.62
Hulless 2row barley	Naked barley	0.20	0.92	0.65	0.84	1.08	1.24
Klock	Oat	1.51	3.81	2.09			
Palu	Oat	1.07	3.32	1.87			
Seger	Oat	1.33	4.64	X			
Spelt wheat Gotland	Ancient	0.70	1.52	X	1.35	2.41	X
Osmo	Oat	2.21	5.57	2.62	0.11	3.32	1.20
Virma	Oat	0.67	1.95	1.51			
Öland	Landrace wheat	0.61	1.52	1.93	0.81	1.71	2.18

X = mineral value lacking 2014.

**Table 7 foods-10-00393-t007:** Means of total yield (kg ha^−1^), protein content (%), Fe and Zn nutritional yield (NY; adults/ha/year) and nutrient density (ND; amount in g needed to be consumed to achieve 100% of the daily recommended intake) of spring cereal groups cultivated in four different locations over three years.

Source	Yield	Protein	Zn NY	Fe NY	Zn ND	Fe ND
Rye	2330 ^b^	11.4 ^b,c,d^	22.3 ^a,b^	32.0 ^a,b^	197 ^b,c^	283 ^b,c^
Oats	2790 ^b^	11.6 ^c,d^	22.8 ^b^	27.8 ^b^	205 ^b^	254 ^c^
Wheat	3060 ^a,b^	12.2 ^a,b,c^	27.9 ^a^	49.8 ^a^	214 ^b^	303 ^b^
Ancient wheat	2830 ^b^	13.3 ^a^	20.9 ^b^	33.8 ^a,b^	175 ^c^	279 ^b,c^
Naked barley	2580 ^b^	13.1 ^a,b^	26.0 ^a,b^	33.3 ^a,b^	218 ^b^	277 ^b,c^
Barley	3550 ^a^	11.0 ^d^	21.5 ^b^	27.3 ^b^	270 ^a^	344 ^a^
Landrace wheat	3040 ^a,b^	12.3 ^a,b,c,d^	30.4 ^a^	41.9 ^a^	201 ^b,c^	277 ^b,c^

Numbers followed by the same letters within a column do not differ significantly by the use of Tukey post-hoc test at *p* < 0.05.

**Table 8 foods-10-00393-t008:** Contribution to four of the most essential minerals in the human diet by 250 g of the product and a comparison with the daily recommended intake (DRI).

Product	Minerals (mg)			
	Zn	Fe	Mg	Cu
Rice [62]	3–11	0.2–7	22–62	0.8–6
Pasta [63]	3–6	9–25	107–145	0.4–0.8
White flour [64]	1–3	2–3	44–65	0.4–0.7
Conventional whole grain wheat [58,59]	4.3–7.3	8–10	224–320	0.9–1.2
Whole grain wheat (present study)	9.6	10	320	1.3
NND mix [present study]	10	11	329	1.3
DRI [65]	8	12	315	0.9

NND mix = 25% rye, 12.5% oats, 12.5% wheat, 12.5% ancient wheat, 25% hulless barley, 12.5% wheat landrace.

## Data Availability

The datasets generated for this study are available on request to the corresponding author.

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
