# Peer review of "Locally Adapted and Organically Grown Landrace and Ancient Spring Cereals—A Unique Source of Minerals in the Human Diet"

_foods, 2021, doi:10.3390/foods10020393_

Round 1

Reviewer 1 Report

The purpose of the work is clear, understandable and important. In parallel with the growth of conscious consumer population, the interest to organic production issues is also growing. It is the social responsibility of science to use the results of credible professional research to clarify the real advantages and disadvantages of organic farming - not only in terms of food production and consumption, but also in terms of economic and ecological impacts.

In the introduction, the authors also try to review the scientific background of the field under study, using the latest literature. I note that large international organizations (e.g. CYMMIT) are also involved in this type of research, and have published results, perhaps it is worth considering these in the literature review.

The experimental materials are derived from well-designed and executed field trials. I note that most of the samples are ancient, landraces and / or old varieties, which could already be indicated in both the title and the abstract, drawing more attention to the content of the article. However, I lack control cultivation conditions from the experimental design. This is because researchers have not studied how these plants behave under non-organic growing conditions. With such an experimental design, it is not possible to justify one of the original objectives, the benefits of organic farming, only to compare the tested samples with each other. I am aware that the work done in this way is also enormous, but perhaps it will be feasible in the future.

The applied analytical procedures (ICP-OES and ICP-MS) and the statistical methods are suitable for achieving the goals.

The interpretation of the results is well supported by the tables and figures that also include detailed statistical data.

In the evaluation describing the development of the mineral content (Chapter 3.1), the letter notation is not clear to me, I suggest to make it clear, or to make a list of the notations used.

Statistical evaluation and interpretation of data is understandable and well supported. In some cases, I find the evaluation of the results too detailed. In evaluating the results, the statements are repeated, in some places too general.

In some cases, there is a typo, please correct it: for example in row 97 egnotypes = `genotypes

Overall, there is no novelty in the article that contains a scientific breakthrough. The conclusion that mineral content also depends on genotypes and environmental factors is a known fact. However, the results obtained are a good complement to the information available so far, especially for the so-called minor cereals from organic farming, which are more common in the Nordic countries, and their ancient, landraces and old varieties. The results demonstrate the generally higher mineral content of oats and some ancient wheat cultivars and the lower mineral content of barley. It draws scientific conclusions based on well-designed and correctly evaluated test results, which can help producers, breeders, researchers and even consumers to find their way.

Author Response

Dear reviewer,

We appreciate your valuable comments and suggestions, and your thorough reading and correction of mistakes in the manuscript. Your comments and suggestions have been taken into consideration accordingly, to improve the paper. We have made every attempt to follow your suggestions and recommendations and all corrections are marked in the manuscript by track changes. Below we provide point-by-point responses to your comments.

The purpose of the work is clear, understandable and important. In parallel with the growth of conscious consumer population, the interest to organic production issues is also growing. It is the social responsibility of science to use the results of credible professional research to clarify the real advantages and disadvantages of organic farming - not only in terms of food production and consumption, but also in terms of economic and ecological impacts.

Thanks for the positive comments.

In the introduction, the authors also try to review the scientific background of the field under study, using the latest literature. I note that large international organizations (e.g. CYMMIT) are also involved in this type of research, and have published results, perhaps it is worth considering these in the literature review.

Thanks for suggestion. We have included additional references including some from the work at CIMMYT.

The experimental materials are derived from well-designed and executed field trials. I note that most of the samples are ancient, landraces and / or old varieties, which could already be indicated in both the title and the abstract, drawing more attention to the content of the article.

Thanks for suggestion. Title and abstract have been changed accordingto suggestion.

However, I lack control cultivation conditions from the experimental design. This is because researchers have not studied how these plants behave under non-organic growing conditions. With such an experimental design, it is not possible to justify one of the original objectives, the benefits of organic farming, only to compare the tested samples with each other. I am aware that the work done in this way is also enormous, but perhaps it will be feasible in the future.

Evaluating genotypes adapted to organic conditions in high-input systems contribute limited information as they do not cope with such environments, similarly as cultivars bred for hig-input systems are not adapted to organic conditions. These issues are now included in teh manuscript.

The applied analytical procedures (ICP-OES and ICP-MS) and the statistical methods are suitable for achieving the goals.

Thanks for positive comments.

The interpretation of the results is well supported by the tables and figures that also include detailed statistical data.

Thanks for positive comments.

In the evaluation describing the development of the mineral content (Chapter 3.1), the letter notation is not clear to me, I suggest to make it clear, or to make a list of the notations used.

Thanks for noting this. Changes have been made for clarity.

Statistical evaluation and interpretation of data is understandable and well supported. In some cases, I find the evaluation of the results too detailed. In evaluating the results, the statements are repeated, in some places too general.

Thanks for pointing that out. The results part have been revised accordingly to decrease repetitions.

In some cases, there is a typo, please correct it: for example in row 97 egnotypes = `genotypes

Thanks for pointing this out. Typos have been corrected.

Overall, there is no novelty in the article that contains a scientific breakthrough. The conclusion that mineral content also depends on genotypes and environmental factors is a known fact. However, the results obtained are a good complement to the information available so far, especially for the so-called minor cereals from organic farming, which are more common in the Nordic countries, and their ancient, landraces and old varieties. The results demonstrate the generally higher mineral content of oats and some ancient wheat cultivars and the lower mineral content of barley. It draws scientific conclusions based on well-designed and correctly evaluated test results, which can help producers, breeders, researchers and even consumers to find their way.

Thanks for this summary and the general positive outcome.

Once again we thank the reviewer for valuable comments and suggestions and hope that with the changes made, the manuscript is now ready for publication in Foods.

Reviewer 2 Report

The subject of this study is very interesting since is linked with the exploitation of organically grown local cereals and landraces. There is needed more information in order to support different claims about their consumption and health benefits. This manuscript represents well-designed research that could bring some interesting insights into the subject.

Although the subject is interesting there are observed some problems in this manuscript. To my opinion, the introduction section should be more specific and report more relevant old and recent findings of the literature. Following are chronologically cited some relevant articles that are not taken into account from the authors.

 (2020). Journal of Food Composition and Analysis, 103660.

 (2020) J Consum Prot Food Saf 15:109–119.

 (2019). Biological trace element research, 187(2), 568-578.

 (2018). Chapter 6 - Trace and Major Elements Content of Cereals and Proteinaceous Feeds in Greece Analyzed by Inductively Coupled Plasma Mass Spectrometry. In A. M. Holban & A. M. Grumezescu (Eds.), Food Quality: Balancing Health and Disease,  (pp. 197-223): Academic Press.

 (2016) Food Addit Contam B 9:261–267

 (2015) Nutr Cycl Agroecosyst 103:347–358

 (2013). Food and Energy Security, 2(2), 81-95.

 (2010) Soil Till Res 107:97–105

Another drawback of the manuscript is the incomplete way of presenting the materials and methods section. Subsection reporting reagents used, sample collection and preparation for the detection are missing. Moreover, information related to the detection of macro and microelements was not reported (Operation conditions, wavelengths, detection limits for each element detected and measured).

The factors that contribute to the two-dimensional model must be shown in one table to understand the discussion.

Also, the references must be reported properly and the keywords must be revised.

Other issues of the manuscript are noted in the pdf file with highlights and comments since part of the manuscript has not numbered lines and thus, I was not able to refer to them when making comments.

Author Response

Dear reviewer,

We appreciate your valuable comments and suggestions given below and in the PDF and your thorough reading and correction of mistakes in the manuscript. Your comments and suggestions have been taken into consideration accordingly, to improve the paper. We have made every attempt to follow your suggestions and recommendations and all corrections are marked in the manuscript by track changes. Below we provide point-by-point responses to your comments and to suggestions in the PDF.

The subject of this study is very interesting since is linked with the exploitation of organically grown local cereals and landraces. There is needed more information in order to support different claims about their consumption and health benefits. This manuscript represents well-designed research that could bring some interesting insights into the subject.

Thanks for your positive feed-back.

Although the subject is interesting there are observed some problems in this manuscript. To my opinion, the introduction section should be more specific and report more relevant old and recent findings of the literature. Following are chronologically cited some relevant articles that are not taken into account from the authors.

Thanks for your promt suggestions of literature to add. The suggested publications together with additional literature from an additional literature search done by ourselves have now been added to the introduction. Furthermore, changes in the text has been done according to the reviewers suggestions to improve the introduction part.

 (2020). Journal of Food Composition and Analysis, 103660.

 (2020) J Consum Prot Food Saf 15:109–119.

 (2019). Biological trace element research, 187(2), 568-578.

 (2018). Chapter 6 - Trace and Major Elements Content of Cereals and Proteinaceous Feeds in Greece Analyzed by Inductively Coupled Plasma Mass Spectrometry. In A. M. Holban & A. M. Grumezescu (Eds.), Food Quality: Balancing Health and Disease,  (pp. 197-223): Academic Press.

 (2016) Food Addit Contam B 9:261–267

 (2015) Nutr Cycl Agroecosyst 103:347–358

 (2013). Food and Energy Security, 2(2), 81-95.

 (2010) Soil Till Res 107:97–105

Another drawback of the manuscript is the incomplete way of presenting the materials and methods section. Subsection reporting reagents used, sample collection and preparation for the detection are missing. Moreover, information related to the detection of macro and microelements was not reported (Operation conditions, wavelengths, detection limits for each element detected and measured).

The materials and methods  section has now been improved in line with suggestions from teh reviewer. We thank the reviewer for the observant suggestions.

The factors that contribute to the two-dimensional model must be shown in one table to understand the discussion.

PCA analyses carried out and reasons for them has been futher elaborated on in teh text now.

Also, the references must be reported properly and the keywords must be revised.

Key words have been revised. References are correct in the submitted file. We will check so that it will also be correct in the version that will be published. Suggested references have been added.

Other issues of the manuscript are noted in the pdf file with highlights and comments since part of the manuscript has not numbered lines and thus, I was not able to refer to them when making comments.

Submitted file has correct numbering. We do not know why the manuscript sent for review has no correct numbering. We have taken all comments in the PDF into consideration and made changes accordingly with few exceptions. Acctually, teh only change we have not done according to the reviewers comments is to add units for each element in the Table 3. This table present mean square values which do not have any units by definition. Because of that, we could not add any units.

Once again we thank the reviewer for valuable comments and suggestions and hope that with the changes made, the manuscript is now ready for publication in Foods.

Reviewer 3 Report

Dear Authors,

In presented study Authors evaluated 25  spring cereal genotypes grown per three years, in four Sweden locations. The minerals content, nutritional yield and nutrient density of the cereals were evaluated. In this paper many results are given, but they only concern the local grown cereals and are not up to date (the cereals were grown in the years 2011-2013). That limits the value and significance of the presented work.

I recommend improved this paper and take major revision.

The pages and the lines throughout the manuscript are not correctly numbered, so the comments are given in the PDF file.

Author Response

Dear reviewer,

We appreciate your recommendation to accept the paper after major revision and also your valuable comments given in the PDF and your thorough reading and correction of mistakes in the manuscript. Your comments and suggestions have been taken into consideration, accordingly, to improve the paper. We have made every attempt to follow your suggestions and recommendations and all corrections are marked in the manuscript by track changes. The few occasions were we have not followed suggestions by the reviewer is pointed out below;

Table 1. The d belongs to the name of the genotype, which is "Spelt wheat Gotland d". In the version of the manuscript that we have, it goes together. We will be careful in checking the last version of the manuscript before publishing so that it is correct in that version.

In section 3.1. We have included Supplementary Table A1 in the manuscript file we submitted. We do not know why it was not sent to the reviewer.

Table 3. Significance levels given are correct.

Table 6. This table needs a full page, which is the reason that it is placed were it is. We expect the publisher to place it in the most appropriate place.

Hereby, we hope that the manuscript is now suitable for publication in Foods.

Round 2

Reviewer 2 Report

I have no further comments.

Reviewer 3 Report

Dear Authors,

The article has been significantly improved. The errors indicated in the previous revision have been corrected. The supplementary materials (Table A1 and Table A2) have been added to text of manuscript.